# Cross-Country Differences and Similarities in Undernutrition Prevalence and Risk as Measured by SCREEN II in Community-Dwelling Older Adults

**DOI:** 10.3390/healthcare8020151

**Published:** 2020-06-02

**Authors:** Jos W. Borkent, Heather Keller, Carol Wham, Fleur Wijers, Marian A. E. de van der Schueren

**Affiliations:** 1Department of Nutrition and Health, School of Allied Health, HAN University of Applied Sciences, 6525 EN Nijmegen, The Netherlands; fleur_trw@hotmail.com (F.W.); marian.devanderschueren@han.nl (M.A.E.d.v.d.S.); 2Schlegel-University of Waterloo Research Institute for Aging, Department of Kinesiology, University of Waterloo, Waterloo, ON N2G 0E2, Canada; hkeller@uwaterloo.ca; 3School of Sport, Exercise and Nutrition, Massey University, Auckland 0632, New Zealand; C.A.Wham@massey.ac.nz

**Keywords:** undernutrition, malnutrition, community-dwelling older adults, nutritional risk, screening tools, risk screening, SCREEN II, self-screening

## Abstract

Undernutrition is highly prevalent among community-dwelling older adults. Early identification of nutrition risk is important to prevent or treat undernutrition. This study describes the prevalence rates of nutrition risk in community-dwelling older adults (aged ≥ 65) using the same validated tool across different countries and aims to identify differences in nutritional risk factors. Cross-sectional data was obtained from three datasets including participants from the Netherlands (NL), Canada (CA) and New Zealand (NZ). Seniors in the Community Risk Evaluation for Eating and Nutrition II (SCREEN II) was used to assess nutritional risk factors and prevalence of risk. Differences between countries were tested with logistic and linear regression. Sensitivity analyses were conducted to test the influence of sampling strategy. A total of 13,340 participants were included, and 66.3% were found to be at high nutrition risk. After stratifying the data for method of data sampling, prevalence rates showed some differences across countries (NL: 61.5%, NZ: 68.2%, CA: 70.1%). Risk factor items that contributed to nutrition risk also differed among countries: NZ and CA participants scored higher for weight change, skipping meals, problems with meal preparation, use of meal replacements, problems with biting and chewing, low fluid intake and problems with doing groceries, as compared to participants in NL. Low intake of fruits and vegetables and meat were more prevalent in NL. In conclusion: nutrition risk is a worldwide, highly prevalent problem among community-dwelling older adults, but risk factors contributing to nutrition risk differ by country.

## 1. Introduction

The World Health Organization (WHO) estimated that the number of people aged over 65 years was approximately 524 million in 2010, comprising 8% of the world’s population [1]. The WHO projected that in 2050 this number will rise towards 1.5 billion people, which will then comprise 16% of the world’s population [1]. The aging world population results in more age-related health problems, one of them being undernutrition [2]. The development of undernutrition in older adults is multifactorial and consists of physiological, social, pathological and economic factors [2,3]. Undernutrition can have several unfavorable consequences, such as loss of independence, poorer physical or mental function, reduced quality of life, increased risk of fragility fractures and mortality [4]. 

In order to identify persons at risk for undernutrition, various screening tools are available (such as: Short Nutritional Assessment Questionnaire 65+ (SNAQ65+), malnutrition universal screening tool (MUST) and mini nutritional assessment (short form) (MNA-SF), which are often based on symptoms of (severe) weight loss or a low body mass index (BMI) [5]. In addition, these tools measure undernutrition when functional decline is likely to be already present. Undernutrition in community-living older people in Western societies is typically a process that starts with impaired food intake, and results in significant changes in body composition and functionality [6]. Relevant screening tools that include early determinants of undernutrition provide a more comprehensive view of risk. The identification of early risk factors contributing to an impaired food intake can give direction to promote health status and to prevent undernutrition. A screening tool that focuses on risk factors that can lead to impaired food intake and eventual undernutrition is the Seniors in the Community Risk Evaluation for Eating and Nutrition II (SCREEN II). This tool is a valid and reliable 17-item instrument that, in contrast to other screening tools, assesses upstream and early determinants that can influence food intake (e.g., difficulty with grocery shopping). Identification of these factors makes it possible to take early preventive measures, at a population or individual level, to prevent the onset of undernutrition [7]. 

Previous research based on SCREEN II showed different prevalence rates of nutrition risk across Western countries. The prevalence of nutrition risk in Dutch community-dwelling adults was 46–67% [8,9], whereas in New Zealand this was 52% [10] and 34% in Canada [11]. However, different cut-off values for SCREEN II (≤53 points vs. ≤49 points), different questionnaires (short form vs. full version of SCREEN II) and different methods of data collection (online questionnaires vs. telephone calls) were used, hindering comparison of prevalence rates across studies. Next to these methodological differences, differences in eating patterns, behaviors and risk factors between countries may also explain differences in prevalence rates. The present study was designed to compare prevalence rates based on SCREEN II between Canada, the Netherlands and New Zealand, applying the same cut-off points, and to study potential differences between risk factors contributing to undernutrition across countries.

## 2. Materials and Methods 

### 2.1. Study Design

Cross-sectional data was used from three different datasets: a Canadian dataset including participants from six continents, a dataset from the Netherlands and one from New Zealand. All datasets contain data on participant age, gender and all seventeen items from the SCREEN II questionnaire. 

### 2.2. Data Collection 

#### 2.2.1. Canadian Derived International Dataset 

The Canadian dataset contains cross-sectional data from 38,361 participants worldwide (44 countries), with the majority of participants (91%) living in Canada. Data was collected in the period 2015–2018 via the website www.nutritionscreen.ca [12]. This online self-screening survey contains the SCREEN II questionnaire. At the beginning of the questionnaire, participants were informed about the possibility that responses could be used for research anonymously. In case of the outcome high nutrition risk, the participant was advised to contact a health professional. From this dataset, only participants of Canada (*n* = 34,822), New Zealand (*n* = 673) and the Netherlands (*n* = 42) were included. Analyses were conducted by authors at the HAN University of Applied Sciences; the HAN University of Applied Sciences Ethical Review Committee judged that no ethical approval was necessary as long as participants were informed about the possibility that answers can be used for research anonymously. 

#### 2.2.2. Dutch Dataset 

The Dutch data was derived in the period April 2017–February 2019 from the website www.goedgevoedouderworden.nl (translated as: “healthy eating for healthy ageing”) [13] and contains cross-sectional data from 4848 participants. This website is based on the international website www.nutritionscreen.ca [12]. The Dutch version was developed by the Dutch Malnutrition Steering Group and gives advice and information about nutrition and physical activity, recipes and contains several health tests. One of these online self-tests is the questionnaire “Hoe eet ik nu?” (translated as: “How do I eat now”)?” based on SCREEN II. The HAN University of Applied Sciences Ethical Review Committee judged that no ethical approval was necessary as long as participants were informed about the possibility that answers can be used for research anonymously. This data is previously described by Borkent et al. [14].

#### 2.2.3. New Zealand Dataset

For this study, data was used from the 2014 wave of the longitudinal cohort, The New Zealand Health, Work and Retirement Study (HART 2014 survey) (*n* = 3050). Participants in this study were approached via electoral rolls on which registration is mandatory in New Zealand. Inclusion criteria was age > 55 years and participants were excluded if they were institutionalized. A postal questionnaire was used to assess individual factors related to retirement, wellbeing and independence. Participants filled in the questionnaire by themselves. Nutrition risk was determined once, in 2014, using the SCREEN II questionnaire. Inclusion and exclusion criteria are described elsewhere [15]. The study was approved by the Massey University Human Ethics Committee. 

#### 2.2.4. Participants Current Study 

All datasets were merged, resulting in data from 46,259 participants. All participants aged < 65 years or who had not completed the full questionnaire were excluded. After excluding participants from countries other than Canada, the Netherlands or New Zealand 13,340 remained for data-analyses. 

### 2.3. Measurements

Nutrition risk and risk factors were measured with the SCREEN II questionnaire. SCREEN II is a 17-item tool that covers the following nutritional risk factors: weight change, perception of body weight, skipping meals, avoidance of products, appetite, intake of dairy/meat (replacements)/fruit and vegetables and fluids, problems with biting and chewing or coughing and swallowing, use of meal replacements, eating with others, who prepares meals (not scored), difficulties with meal preparation, and problems with doing groceries. Each response was given a score from 0 to 4, where a score ≤2 is characterized as a potential nutrition risk for the specific item [7]. The sum of the 16 scored items results in a total score of ranging from 0 to 64, with a lower score indicating a higher nutrition risk. According to Keller et al. [7] a total score ≤53 is indicative of any nutrition risk and ≤49 as high nutrition risk. Information about age, gender and country was measured in each data set and was used to characterize the samples.

### 2.4. Data Analysis

Continuous data was checked for normality by stem and leaf plots and QQ-plots. Descriptive statistics (means with standard deviation and number with frequencies) were used to represent the characteristics of the participants, prevalence rates of scored items of the SCREEN II questionnaire and the total score for nutrition risk. Logistic regression analyses were applied to test associations between countries (independent variable) and nutrition risk (dependent variable), based on the cut-off values for nutrition risk (≤53) and high nutrition risk (≤49) [7]. Logistic regression analyses were also used to assess the associations between countries (independent variable) and separate items of SCREEN II (nutritional risk factors) as dependent variables. For this analysis, the score for every single SCREEN II item was dichotomized into potentially leading to nutrition risk (≤2) and not leading to risk (≥3) [7]. Lastly, a linear regression analysis was performed to assess the association between the independent variable ‘country’ and the total SCREEN II score as the dependent variable. The Netherlands was used as the reference category in all analyses. Previous studies showed associations between nutrition risk and age categories [11,14] and between nutrition risk and gender [11,16]. Therefore, these variables were added as potential confounders to models. Statistical analyses were performed using SPSS V24.0 (IBM, Chicago VS); because of the large sample-size, values were considered significant when *p* < 0.01.

#### Sensitivity Analysis 

Sensitivity analyses were performed because of the differences in recruitment of participants between datasets; online sampling in Canada and the Netherlands, vs. sampling by electoral roll in New Zealand. Datasets derived from online surveys were also analyzed without the New Zealand dataset. 

## 3. Results

The total number of participants in the different datasets was 46,259 (Figure 1). A total of 32,843 participants were excluded, because they did not meet the inclusion criteria (age < 65, *n* = 30,769; incomplete questionnaires, *n* = 1508; countries other than Canada, the Netherlands or New Zealand *n* = 642). This resulted in a total population of 13,340 participants to be analyzed. 

The main characteristics of the participants and the total scores of the SCREEN II questionnaire are shown in Table 1. Most participants were from Canada (*n* = 9538; 71.5%), followed by the Netherlands (*n* = 2482; 18.6%) and New Zealand (*n* = 1320; 9.9%). The majority (*n* = 9.796; 73.4%) of the included participants were woman. More than half of the participants (*n* = 8773; 65.8%) were in the younger age category of 65–74 years of age. Overall, the mean total score for SCREEN II was 44.6 (SD: 9.3) with a range from 0 to 64. The overall prevalence of any nutrition risk (total score ≤ 53) was 85.7% (*n* = 11,431) and the prevalence of high nutrition risk (total score ≤ 49) was 66.3% (*n* = 8841). 

### 3.1. Total Score with All Included Datasets 

Nutrition risk was the highest in Canada. Compared to the Netherlands (reference), the odds ratio of being at nutrition risk (total score ≤ 53) in Canada was 1.39 (99%CI: 1.18–1.64) and the odds ratio of being at high nutrition risk (total score ≤ 49) was 1.53 (99%CI: 1.36–1.73). New Zealand had the lowest prevalence with odds ratios of 0.60 (99%CI: 0.48–0.75) and 0.66 (99%CI: 0.55–0.79) respectively. 

### 3.2. Sensitivity Analyses: Total Score with only Datasets from Online Websites

Table 2 shows the main characteristics of the participants and the total scores of the SCREEN II questionnaire when only including data derived through the Canadian and the Dutch websites (*n* = 12,237). The majority of the participants lived in Canada (*n* = 9538, 77.9%), the Netherlands (*n* = 2.482, 20.3%), and a relatively small group that lived in New Zealand (*n* = 217, 1.8%) was obtained from the Canadian dataset. Two-thirds of the participants (*n* = 7902; 64.6%) were in the younger age category of 65–74 years of age and most of the participants (*n* = 9193; 75.1%) were woman. The mean total score for nutritional risk from New Zealand decreased by 5.7 points, compared to the data set described in Table 1. This resulted in a 1.17 (99%CI: 0.70–1.96) times higher odds of being at any nutrition risk (total score ≤ 53) and a 1.47 (99%CI: 1.00–2.18) times higher odds of being at high nutrition risk (total score ≤ 49) in New Zealand as compared to the Netherlands. 

As prevalence rates of scored nutritional risk factors changed after stratifying by the source of data collection, Table 3 presents the results of the datasets from online questionnaires only, to be able to make a true comparison of nutritional risk factors among countries. The results of the differences in risk factors between countries with all data are presented in supplementary Table A1 (Appendix A). 

### 3.3. Nutritional Risk Items

Overall, the most frequently reported nutritional risk factors on the SCREEN II questionnaire were: problems with perception on bodyweight (72.4%), low intake of fruit and vegetables (52.7%)/meat, eggs, fish or meat substitute (41.8%)/dairy (60.8%), limitation and avoiding of certain foods (50.2%), problems with meal preparation (50.0%), change in bodyweight (49.2%) and eating alone (42.0%).

Significant differences were seen in most items between the Netherlands and New Zealand, except in perception of bodyweight, limitation or avoiding certain products, intake of meat and alternatives, poor appetite, low fluid intake and problems with coughing. Significant differences between the Netherlands and Canada were seen in all nutritional risk factors, except in problems with appetite, problems with coughing and eating meals with others. Overall, nutritional risk factors were reported more frequently in New Zealand and Canada compared to the Netherlands, with the exception of the intake of fruit and vegetables, and meat and alternatives. These risk factors were more frequent in the Netherlands.

## 4. Discussion

This study showed different prevalence rates for any nutrition risk according to SCREEN II (total score ≤53) in Canada (87.7%), the Netherlands (84.1%) and New Zealand (74.0%). However, these differences were attenuated after stratifying the data by the source of the data collection: the prevalence rates of any nutrition risk (total score ≤53) for data obtained only from online-self administration was 87.7% (Canada), 84.1% (the Netherlands) and 85.3% (New Zealand). In contrast with the total score for nutrition risk, notable differences in single items of the SCREEN II questionnaire were shown. These findings are in line with our hypothesis that differences in food consumption, habits and sociocultural status between countries result in a varying prevalence of factors that contribute to nutrition risk. As a result of these differences, nutritional interventions should focus on frequently scored risk factors in specific countries, and should be tailored to specific age groups [14].

Previous studies showed different prevalence rates between countries; the prevalence rate of nutrition risk according to the SCREEN II was 46–67% in the Netherlands [8,9], 52% in New Zealand [10] and 34% in Canada [11]. As described earlier, methodological issues may have contributed to these differences. The current study showed higher prevalence rates of nutrition risk, but only small differences between countries. The higher prevalence could be explained by the method of sampling. The majority of the participants in this study were recruited via online websites that provided information about healthy aging. These websites are thought to be used by health care professionals to refer older adults who may be at increased risk of undernutrition. Alternatively, older adults who are concerned about their nutritional health may seek out these websites. However, this selection bias is likely to occur in each country (Canada, the Netherlands and New Zealand) and presumed to not vary by country of origin. Selection bias mainly affects prevalence rates [17] but associations are relatively immune for this type of bias [17,18,19]. So, despite the possibility of a non-representative study-population, the comparison of individual nutritional risk factors provides a valuable insight in differences between three western countries.

The data used for this study is from three different continents. However, this data is likely not to be representative for other countries, especially in North America and Europe. Canada and the Netherlands are rich countries with a well-developed health care system and good social securities. It is likely that other countries in Europe and North America have other nutritional risk factors for malnutrition. No data is available for such countries, and a validation study is needed to determine if SCREEN II is applicable in less developed countries and other continents.

SCREEN II is originally designed as a nutrition screening tool [7]. Therefore, it does not address the full spectrum of psychological and physical determinants associated with undernutrition. Two questions addressing these determinants are ‘eating alone’ and ‘inability to perform grocery shopping’, but we do not have data to test how these two questions relate to psychological or physical health. A recent publication showed that SCREEN II was associated with food intake but only in a lesser extent to physical parameters, while the SNAQ^65+^ screening tool was more related to physical health [20]. The combination of the two might provide supplementary information on undernutrition risk, covering both early stage and late stage malnutrition [20].

To our knowledge, this is the first study exploring nutritional risk factors across three countries. Differences were seen in the intake of several food groups. The majority of the Dutch participants (65.9%) consumed three or less portions of fruit or vegetables per day. This risk factor was less commonly reported in New Zealand (39.2%) and Canada (49.2%); however, this difference may be explained by potatoes being considered a vegetable in the latter countries [21,22]. A low intake of meat and alternatives was frequently reported by participants from the Netherlands (55.4%) and New Zealand (57.1%), but to a lesser extent in Canada (37.9%). The higher meat intake in Canada is line with the national food consumption surveys, where the meat consumption of older adults in the Netherlands (men: 134 g per day; women: 114 g per day) [23] was lower than the meat consumption of older adults in Canada (men: 189 g per day; women: 140 g per day) [24]. In contrast, dairy consumption was higher in the Netherlands, with 44.7% using more than one serving of dairy per day, compared to New Zealand (34.6%) and Canada (37.9%). Both meat and alternatives and dairy products are a major source of protein intake in the Netherlands [23], Canada [24] and New Zealand [25]. Protein intake is associated with changes in lean body mass in community-dwelling older adults [26], and increasing the intake of dairy and meat (or alternatives) is therefore an important public health message targeting this age group. 

Between-country differences were also seen in fluid intake. The prevalence rate of this risk factor was the highest in Canada (40.1%) and lowest in the Netherlands (25.1%). Although fluid intake not only consists of water intake, differences in fluid use between countries are difficult to explain. Seasonal differences could have played a role, but this is unlikely as the surveys were completed throughout the year in all of the data collections. Fluid intake decreases with age and older adults are therefore at higher risk of dehydration [27]. However, no differences in proportions in different age groups were seen in this analysis. Attention to intake of fluid can prevent dehydration [28], which is more prevalent in older adults [29]. 

Differences were also seen in problems with biting or chewing. This risk factor was scored more frequently in New Zealand (20.3%) and Canada (18.6%), compared to the Netherlands (14.9%). The large differences in biting and chewing problems between the countries have also been shown in a previous study [30]. Good dental care is essential for preventing biting and chewing problems [31]. However, access to dental care differs between countries and is not always guaranteed for older adults with a low income [32]. Biting and chewing problems can result in the avoidance of certain food groups, especially hard fruits, vegetables and meat [33,34]. Processing and cooking can soften fruits and vegetables if needed. Different cooking technics, such as marinating, slow cooking or the use of cooking bags, could make meat easier to chew [35]. 

Finally, differences were seen in problems with doing grocery shopping. This risk factor was scored more frequently in New Zealand (28.1%) and Canada (24.1%), compared to the Netherlands (17.4%). Reasons for the difference may be the distance to the nearest shops, many of those living in the Netherlands can walk or take public transport to grocery shops, whereas older people in NZ are reliant on the use of a private car, as public transport is limited, and in Canada, the winter season will affect access to grocery stores for many older adults. Possible difficulties with traveling to the supermarket and the inability to reach items, to push trolleys and carry groceries home, especially by public transport, could hinder older adults in buying some food items [36,37]. Interventions could focus on support from relatives or friends, to assist with grocery shopping or promoting delivery services and online shopping. 

One of the strengths of this study is the large sample size. Pooling dataset from three different countries/continents made it possible to do meaningful analyses on overall nutrition risk and risk factors contributing to risk. Another strength is the use of different forms of sampling, and when sensitivity analyses were completed, the differences in prevalence due to sampling methods could be detected. These findings suggest that the sampling method may be more important than countries when determining the prevalence of nutrition risk. 

A major limitation in this study is the lack of background information of the participants, specifically from the online version of SCREEN II. Only gender and age, important confounders [11,14], were available and, thus, further characterization of the sample, as well as control for potential confounding was not possible. Other relevant characteristics to measure in future studies should include educational status, living status and comorbidities. Secondly, selection bias is a potential limitation. The Canadian and Dutch websites (www.nutritionscreen.ca and www.goedgevoedouderworden.nl [12,13]) were widely available, and participants may have had a special interest in nutrition or may have had a specific concern, that may have resulted in a higher prevalence of nutrition risk. Internet access is high in the Netherlands, Canada and New Zealand (>75% in adults aged 65+) [38,39,40], but especially older adults with a higher social economic status and women are more likely to search for health information [41]. This likely resulted in selection bias. As stated previously, these types of selection bias are likely to be non-differential between countries and mainly affect prevalence rates but not associations. In addition, as a consequence of the use of online questionnaires, participants could have filled in the questionnaires several times, and/or provided incorrect answers, which could not be verified.

## 5. Conclusions

In conclusion, nutrition risk is a worldwide common problem among community-dwelling older adults, with the majority of older adults displaying multiple nutrition risk factors and/or increased nutritional risk. When controlling for the source of the data, the prevalence of nutrition risk appeared to be quite similar across Canada, the Netherlands and New Zealand. It is important to realize that the prevalence rates of nutrition risk may depend on the method of data collection. This study revealed significant differences between countries in nutrition risk factors, indicating that interventions should differ for each country and should focus on frequently scored risk factors in a specific region. 

## Figures and Tables

**Figure 1 healthcare-08-00151-f001:**
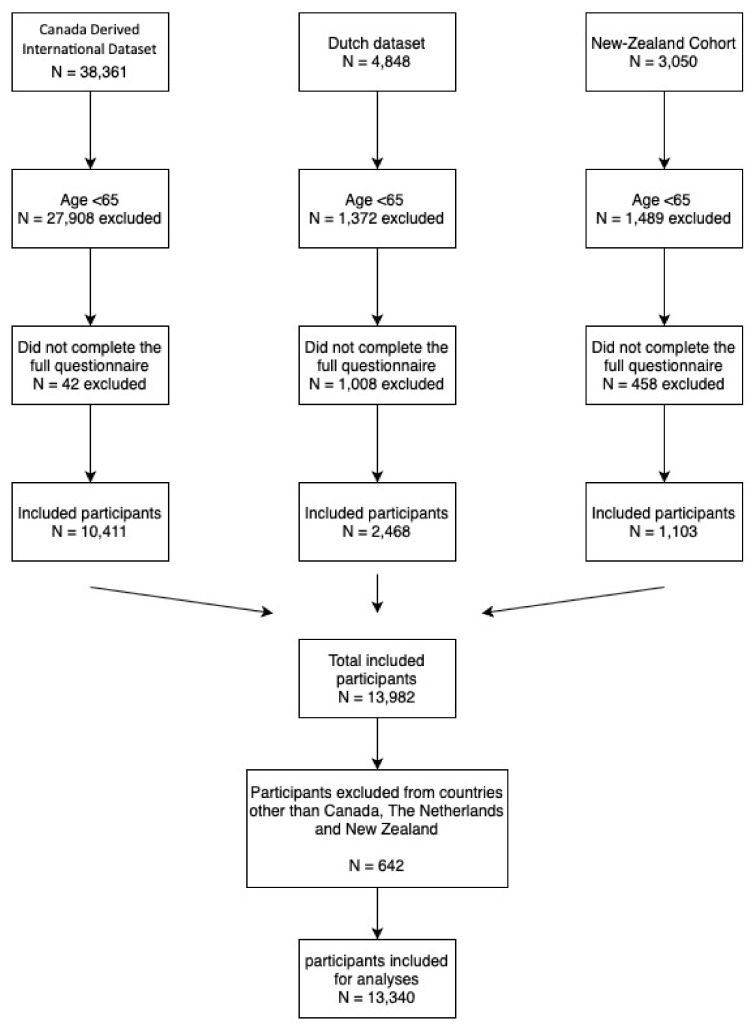
Flowchart of included participants for analyses showed for three datasets.

**Table 1 healthcare-08-00151-t001:** Participant characteristics and Seniors in the Community Risk Evaluation for Eating and Nutrition II (SCREEN II) scores by country/continents.

	Total	The Netherlands	New Zealand	Canada
	*n* = 13,340	*n* = 2482	*n* = 1320	*n* = 9538
**Gender**				
Men	3544 (26.6%)	607 (24.5%)	589 (42.2%)	2368 (24.8%)
Women	9796 (73.4%)	1875 (75.5%)	807 (57.8%)	7170 (75.2%)
**Age**				
65–74	8773 (65.8%)	1437(57.9%)	1025 (77.7%)	6311 (66.2%)
75–84	3569 (26.8%)	805 (32.4%)	283 (21.4%)	2481 (26.0%)
≥85	998 (7.5%)	240 (9.7%)	12 (0.9%)	746 (7.8%)
**SCREEN II**				
≥54	1909 (14.3%)	394 (15.9%)	343 (26.0%)	1172 (12.3%)
≤53 *	11,431 (85.7%)	2088 (84.1%)	977 (74.0%)	8366 (87.7%)
OR		ref	**0.54 (0.43–0.67)**	**1.35 (1.15–1.59)**
Adjusted OR ***			**0.60 (0.48–0.75)**	**1.39 (1.18–1.64)**
**SCREEN II**				
≥50	4499 (33.7%)	956 (38.5%)	690 (52.3%)	2853 (29.9%)
≤49 **	8841 (66.3%)	1526 (61.5%)	630 (47.7%)	6685 (70.1%)
OR		ref	**0.57 (0.48–0.68)**	**1.47 (1.30–1.66)**
Adjusted OR ***			**0.66 (0.55–0.79)**	**1.53 (1.36–1.73)**
**SCREEN II**				
total score	44.6 (9.3)	46.1 (8.3)	48.7 (7.4)	43.7 (9.5)
Regression coefficient		ref	**2.6 (1.9; 3.5)**	**−2.4 (−2.9; −1.8)**
Adjusted Regression coefficient ***			**1.4 (0.6; 2.2)**	**−2.7 (−3.2; −2.2)**

Data are shown as number (percentage)/mean (standard deviation) or OR (99%CI)/Mean differences (99%CI). * indicative for nutrition risk according to Keller et al.; ** indicative for high nutrition risk according to Keller et al. *** adjusted for age and gender. Bold = significantly different from reference category.

**Table 2 healthcare-08-00151-t002:** Characteristics of the participants and total scores of SCREEN II questionnaire, compared between countries with data only from online websites.

	Total	The Netherlands	New Zealand	Canada
	*n* = 12,237	*n* = 2482	*n* = 217	*n* = 9538
**Gender**				
Men	3044 (24.9%)	607 (24.5%)	69 (31.8%)	2368 (24.8%)
Women	9193 (75.1%)	1875 (75.5%)	148 (68.2%)	7170 (75.2%)
**Age**				
65–74	7902 (64.6%)	1437 (57.9%)	154 (71.0%)	6311 (66.2%)
75–84	3340 (27.3%)	805 (32.4%)	54 (24.9%)	2481 (26.0%)
≥85	995 (8.1%)	240 (9.7%)	9 (4.1%)	746 (7.8%)
**SCREEN II**				
≥54	1598(13.1%)	394 (15.9%)	32 (14.7%)	1172 (12.3%)
≤53 *	10,639 (86.9%)	2088 (84.1%)	185 (85.3%)	8366 (87.7%)
OR		ref	1.09 (0.65; 1.82)	**1.35 (1.15; 1.59)**
Adjusted OR ***			1.17 (0.70; 1.96)	**1.39 (1.18; 1.64)**
**SCREEN II**				
≥50	3878 (31.7%)	956 (38.5%)	69 (31.8%)	2853 (29.9%)
≤49 **	8359 (68.3%)	1526 (61.5%)	148 (68.2%)	6685 (70.1%)
OR		ref	1.34 (0.91; 1.99)	**1.47 (1.30; 1.66)**
Adjusted OR ***			1.47 (1.00; 2.18)	**1.54 (1.36; 1.74)**
**SCREEN II**				
total score	44.2 (9.3)	46.1 (8.3)	43.0 (10.3)	43.7 (9.5)
Regression coefficient		ref	**−3.1 (−4.8; −1.4)**	**−2.4 (−2.9; −1.8)**
Adjusted Regression coefficient ***			**−3.8 (−5.5; −2.2)**	**−2.7 (−3.2; −2.2)**

Data are shown as number (percentage)/mean (standard deviation) or OR (99%CI)/Mean differences (99%CI). * indicative of nutrition risk according to Keller et al. ** indicative of high nutrition risk according to Keller et al. *** adjusted for age and gender. Bold = significantly different from reference category.

**Table 3 healthcare-08-00151-t003:** Nutritional risk factors according to the SCREEN II compared between countries, with data from online websites.

	Total	The Netherlands	New Zealand	Canada
	*n* = 12,237	*n* = 2482	*n* = 293	*n* = 9538
**Change in weight in last six months** (gained or lost ≥ 2.5 kg)	6023 (49.2%)	964 (38.8%)	119 (54.8%)	4940 (51.8%)
OR		ref	**1.91 (1.33–2.76)**	**1.59 (1.31–1.92)**
adjusted OR *			**2.00 (1.38–2.89)**	**1.73 (1.43–2.10)**
**Unintentional weight change last six months**	1703 (13.9%)	245 (9.9%)	45 (20.7%)	1413 (14.8%)
OR		ref	**2.39 (1.50–3.80)**	**1.48 (1.24–1.76)**
adjusted OR *			**2.87 (1.79–4.61)**	**1.62 (1.35–1.93)**
**Perception bodyweight** (more or less than it should be)	8864 (72.4%)	1554 (62.6%)	145 (66.8%)	7165 (75.1%)
OR		ref	1.20 (0.82–1.77)	**1.80 (1.59–2.04)**
adjusted OR **			1.13 (0.77–1.67)	**1.74 (1.54–1.97)**
**Skipping meals** (sometimes or more frequent)	4632 (37.9%)	536 (21.6%)	64 (29.5%)	4032 (42.3%)
OR		ref	**1.52 (1.01–2.27)**	**2.66 (2.32–3.05)**
adjusted OR *			**1.62 (1.08–2.42)**	**2.76 (2.40–3.16)**
**Limitation or avoiding certain products**	6148 (50.2%)	1011 (40.7%)	100 (46.1%)	5037 (52.8%)
OR		ref	1.24 (0.86–1.79)	**1.63 (1.45–1.83)**
adjusted OR *			1.27 (0.88– 1.83)	**1.63 (1.45–1.84)**
**Fair/poor Appetite**	2595 (21.2%)	597 (24.1%)	55 (25.3%)	1943 (20.4%)
OR		ref	1.07 (0.70–1.63)	**0.81 (0.70–0.93)**
adjusted OR *			1.35 (0.87–2.08)	0.88 (0.76–1.02)
**Low intake fruit or vegetables per day** (three or less portions)	6455 (52.7%)	1674 (67.4%)	85 (39.2%)	4696 (49.2%)
OR		ref	**0.31 (0.21–0.45)**	**0.38 (0.28–0.52)**
adjusted OR *			**0.32 (0.22–0.46)**	**0.48 (0.42–0.54)**
**Low intake of meat, eggs, fish or meat substitute** (once a day or less)	5113 (41.8%)	1376 (55.4%)	124 (57.1%)	3613 (37.9%)
OR		ref	1.07 (0.74–1.55)	**0.49 (0.44–0.55)**
adjusted OR *			1.12 (0.77–1.62)	**0.50 (0.45–0.56)**
**Low dairy intake** (one portion a day or less)	7434 (60.8%)	1373 (55.3%)	142 (65.4%)	5919 (62.1%)
OR		ref	**1.53 (1.04–2.24)**	**1.32 (1.18–1.49)**
adjusted OR *			**1.51 (1.03–2.22)**	**1.32 (1.17–1.48)**
**Low fluid intake** (≤four glasses)	4514 (36.9%)	622 (25.1%)	65 (30.0%)	3827 (40.1%)
OR		ref	1.28 (0.86–1.91)	**2.00 (1.76–2.28)**
adjusted OR *			1.41 (0.94–2.11)	**2.16 (1.89–2.46)**
**Problems with coughing, choking or pain when swallowing** (sometimes or often)	2245 (18.3%)	437 (17.6%)	37 (17.1%)	1771 (18.6%)
OR		ref	0.96 (0.59–1.56)	1.07 (0.92–1.24)
adjusted OR *			1.08 (0.66–1.77)	1.12 (0.96–1.31)
**Problems with biting or chewing** (sometimes or often)	2187 (17.9%)	369 (14.9%)	44 (20.3%)	1774 (18.6%)
OR		ref	1.46 (0.92–2.30)	**1.31 (1.12–1.54)**
adjusted OR *			**1.78 (1.12–2.85)**	**1.44 (1.22–1.70)**
**Use of meal replacements/supplements** (sometimes or more frequent)	2362 (19.3%)	236 (9.5%)	31 (14.3%)	2095 (22.0%)
OR		ref	**1.59 (0.93–2.69)**	**2.68 (2.22–3.23)**
adjusted OR *			**1.78 (1.04– 3.03)**	**2.89 (2.39–3.49)**
**Eating meals with others** (sometimes or fewer)	5143 (42.0%)	1014 (40.9%)	125 (57.6%)	4004 (42.0%)
OR		ref	**1.97 (1.36– 2.85)**	1.05 (0.93–1.18)
adjusted OR *			**2.31 (1.58–3.35)**	1.12 (0.99–1.26)
**Meal preparation** (meal preparation is hard/I do not enjoy the meals that were prepared for me)	6123 (50.0%)	984 (39.6%)	135 (62.2%)	5004 (52.5%)
OR		ref	**2.51 (1.72–3.65)**	**1.68 (1.49–1.89)**
adjusted OR *			**2.** **94 (2.00–4.** **33)**	**1.77 (1.57–2.00)**
**Problems with doing groceries** (sometimes or more often)	2794 (22.8%)	433 (17.4%)	61 (28.1%)	2300 (24.1%)
OR		ref	**1.85 (1.22–2.79)**	**1.50 (1.30–1.75)**
adjusted OR *			**2.42 (1.58–3.71)**	**1.72 (1.48–2.01)**

Note: data is presented as number (percentage) or OR (99%CI) * adjusted for age and gender. Bold = significantly different from reference category.

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
