# Peer review of "Cross-Country Differences and Similarities in Undernutrition Prevalence and Risk as Measured by SCREEN II in Community-Dwelling Older Adults"

_healthcare, 2020, doi:10.3390/healthcare8020151_

Round 1

Reviewer 1 Report

Thank you for the opportunity to review this valuable manuscript. With the rising proportion of older adults across the world, assessing nutrition risk is increasingly pertinent, and understanding that risk factors differ across countries aids public health responses in improving older adult health outcomes.

The manuscript is well-written and structured.

In the title, cross-continent differences – the Netherlands is not a continent. The authors do not claim that the Netherlands is representative of Europe, consideration to alter the ‘continent’ wording recommended.

There is no justification as to why Australia was included, to create a generalised “Australiasian” cohort in conjunction with New Zealand. The data was taken from the Canadian dataset, whereas Netherlands and New Zealand had administered and collected data within their own countries. The number of Australian participants appears immaterial, in the Canadian dataset (n=242) and in final analysis (n=76), which makes up just 5% of the “Australiasian” cohort. The conclusions of this study imply the nutritional risk factors for Australia and New Zealand are the same, however 95% of these apply to New Zealand. Although there are many similarities between these countries largely based on geographical proximity, factors such as ethnicity, pension entitlements and healthcare support are different and result in substantial differences for the older adult population. It is recommended that this manuscript be presented for cross-country differences between Canada, Netherlands and New Zealand.

Granted, the Netherlands cohort in the Canadian dataset was lower (n=42), however the large dataset from the Netherlands website justifies inclusion of this country and data.

In 2.2.3, the recruitment method is detailed, however this is not outlined for the Canadian and Dutch datasets.

Appropriate analyses were carried out, taking into account the different data collection methods. Justification for use of Netherlands as the reference in regression analyses was not provided.

In the paragraph starting 218, selection was well addressed. In addition, consideration of the higher literacy of these older adults required to complete online surveys would strengthen this area, as well as equity in access to technology.

Lines 239-246 and 266 are inconsistent with “Australiasia”, as they discuss the situation in New Zealand only. These are accurate observations – need to include Australia specific references as well if Australia is to be included in this manuscript.

Line 247 – unclear as to why ‘meat alternatives’ are included in the recommendations – this should be Meat & alternatives

Line 253 – consideration of season surveyed not displayed, which will impact fluid consumption

Line 269 – authors may consider the weight and bulk of purchases on public transport or when shopping on foot as limiting food choices.

Author Response

Thank you for reviewing our article. Please see attachment for our point-to-point reply.

Reviewer 2 Report

Thank you for the opportunity to review this paper. The topic on the under-nutrition of older people would be significant globally. This study used the online data set. Even though the authors presented the limitation of the study such as lack of demographics, possible confounding variables such as co-morbidities, living arrangement and community environments (accessible location to groceries). Thus, there are limited to generalize the purpose of the study.

Title: Overall, introduction focused on the risk of undernutrition among elderly. The nutrition risk can be included both over and under-nutrition. Thus, the authors should change based on the purpose of the study. Importantly, this study included only the nutritional risk factors excluding psycho-social and physical factors. Thus, the authors should consider the need to change the title.

Introduction: This study included three countries among Western countries. As Healthcare is an International Journal, the authors should explain obviously why three countries should be included among Western countries. I don’t think that the reasons on the different cut-off and prevalence is reasonable for the aims. Namely, Do three countries have more worse undernutrition among older adults than Other countries including Asian countries?

The authors explained that three countries had different cut-off of SCREEN Ⅱ. I think this difference would be induced from different culture including lifestyles.

Accordingly, the authors should explain why the different cut-off of SCREEN Ⅱ in three countries is research problem based on the latest references and reasonable evidence.

Methods: Is age only exclusion criteria for the sampling ?. Most importantly, the data collection period had big gap between three continent. Namely, data from New Zealand  was conducted since 2006. It would not be same condition to compare the data between three countries.

Discussion: The authors explained the nutritional risk factors based on the results of the study. However, the other risk factors including physical or psychological factors of the undernutrition should be discussed compared to nutritional risk factors. Furthermore, the authors should add the difference between young and older people based on the nutritional risk factors to design tailored education program for older people.

References: References should be updated, At present, around one-third was more than 10 years.

I hope this comment would be

helpful to improve the quality of the paper.

Author Response

(The authors gave the same response as above.)

Reviewer 3 Report

In this original research manuscript, the authors explored prevalence rates of nutrition risk in community-dwelling older adults through a validated tool (i.e., SCREEN II) across different continents.

Overall it is a nice manuscript, I have some small comments to do. The authors should be able to address these points for full consideration.

Specific comments

Page 1; Line 30: Please remove numbers before each keyword.

Page 1; Line 39: Pathological factors are still missing. I suggest mentioning it. Proposed reference: Azzolino D et al. Nutrients 2019, 11, 2898; doi:10.3390/nu11122898

Page 2; Line 42: “various screening tools are available”. Which are they? I suggest adding some examples of these screening tools.

Page 2; Line 43: “These tools merely identify people already undernourished.” That's not true at all. Just to date, the Mini Nutritional Assessment (MNA) is a validated tool that identify older persons who are at risk for malnutrition, or who are already malnourished. (Guigoz Y, Vellas J, Garry P. Facts Res Gerontol 1994, 4 (supp. 2):15-59)

Page 2; Line 44-47: This sentence is too long. Please rephrase as does not flow well.

Page 3; line 123: “Cut-off values for nutrition risk (≤53 vs. ≥54) and high nutrition risk (≤49 vs. ≥50).” I suggest removing ≥54 and ≥50 or to explain what the cut-offs ≥54 and ≥50 indicate (i.e., ≥54 no nutrition risk and ≥50 no high nutrition risk)

Page 5; Line 52: “Overall, the mean total score for SCREEN II was 44.6 (SD: 9.3) with a range from 0 to 64.” Do you think that is possible to score 0 at the SCREEN II or simply the score of 0 indicates that the questionnaire has not been completed? Please explain it.

Page 5; Line 170-172: Do you think that the small sample from Australasia (particularly that of Australia) is representative of the population compared to the other countries included in the study?

Page 7; Line 211: Please add the word “only” before “from online-self”

Page 8; 220: Please change “As described in the introduction of this manuscript” to “As described above”

Page 8; 233: Please change “contrasting” to “exploring” (Please check that your intended meaning has not changed).

Page 8; Line 260: “Biting and chewing problems can result in avoidance of certain food groups, especially hard fruits and vegetables.” I suggest adding also “meats” which is one of the major sources of proteins. Proposed reference: Nutrients 2019, 11, 2898; doi:10.3390/nu11122898

Page 9; line 281: Please add “should” before “include”

Page 9; Line 282-284: Please rephrase as does not flow well.

Author Response

(The authors gave the same response as above.)

Reviewer 4 Report

the article "Cross- continent differences and similarities in prevalence and risk factors for nutrition risk in community-dwelling older adults." presents a generally well written study. It requires some better minors to be published in Healthcare:

-Table 1 and 2 have much of the same information, please avoid duplication.

-The most striking feature of the article are the Table 3 results: can it be presented in the form of 1 or 2 bar figures or another graphic option that is more attractive?

-Please try to mention or correlate in any way the current and past results with the income level of the countries under study, the above because the results will surely be very different in those countries with middle or low income.

Author Response

(The authors gave the same response as above.)

Round 2

Reviewer 1 Report

All review comments have been appropriately addressed, and manuscript changes made.

Author Response

Thank you for your time and effort for reviewing our article. As you made no additional suggestions, we did not change our manuscript.

Reviewer 2 Report

Thank you for your hard work.

I think this revised paper was well done in a more detail.

I have two comments the authors should consider to change as follow.

  1. However, the amended title is too long, I suggest the title : .... undernutrition prevalence and its risk in .....
  2. In addition, in terms of “gender”, gender should be categorized into “men” and “women” instead of biological term [sex] with male and female.

Thanks.

Author Response

Thank you again for your time and effort in reviewing our article. 

For the title we would prefer to keep it as it is. Shortening it would put a heavy focus on undernutrition, neglecting the risk factors and not mentioning SCREEN.

I hope you understand our reasons for not changing the titles.

We have made the other edits as suggested.